# Culture of Safety and Preparedness: Benefits of Applying a Daily Mitigation Mindset in the Hospital Setting

**DOI:** 10.3390/ijerph21020221

**Published:** 2024-02-13

**Authors:** Anne Reid Griffin, Aram Dobalian, Joanne C. Langan, Sallie J. Shipman

**Affiliations:** 1US Department of Veterans Affairs, Veterans Emergency Management Evaluation Center (VEMEC), North Hills, CA 91343, USA; 2Division of Health Services Management and Policy, College of Public Health, The Ohio State University, Columbus, OH 43210, USA; 3Valentine School of Nursing, Saint Louis University, St. Louis, MO 63104-1099, USA; 4School of Nursing, University of West Florida, Pensacola, FL 32603, USA

**Keywords:** risk evaluation and mitigation, nursing, natural disasters, climate, disaster nursing, disaster management

## Abstract

Aim: Utilizing the subjective experience of nurse executives who have supervised nurses during a major disaster in a hospital setting, this study aims to describe the subjective experience of nurse executives (NE) who have supervised nurses’ responses to major disasters. This paper will focus on strategies to support nursing response to disasters, specifically to strengthen resiliency and the ability to maintain function despite the shock of a disaster, including those caused by climate change. Methods: Semi-structured interviews were conducted with 11 hospital-based nurse executives who supervised nurses during some of the worst natural and human-made disasters in different regions of the United States. A phenomenological approach was used to analyze and describe emerging themes from the qualitative data. Results: Three nurse executives demonstrated theme saturation for mitigation steps to augment traditional disaster readiness activities: (1) Assessment of human infrastructure: daily skills needed during disaster response; (2) Identification and study of failure points and metrics; (3) Strengthening human infrastructure: rectifying deficiencies; and (4) Monitoring metrics and making corrections during conventional periods. Conclusion: Mitigation steps may improve outcomes in hospital function during conventional times; therefore, these steps may improve resiliency and the ability to maintain functions during major disasters, including climate change.

## 1. Introduction

As the largest component of a hospital workforce [1], nurses are the first line of defense and best positioned to maintain function and safety in the face of a disaster event. Nurses represent the hospital human infrastructure, a valuable defense against the health impacts of the worst disasters.

### 1.1. Climate Change and Weather Events

Changes in climate affect the frequency of severe weather events and can result in threats to human health [2,3]. An upward trend in such climate events has occurred within the U.S. and globally between 2018 and 2022 [4]. According to ASPR TRACIE [5], any healthcare agency leaders and emergency management partners should increase their organizational resiliency by incorporating climate change impacts into their emergency preparedness and response initiatives. Hospital resilience has been described as the ability to maintain functioning despite sustaining a shock such as an earthquake, and it is achieved when many different infrastructure contexts (e.g., learning from previous disasters, managing the availability of staff, daily communication practices) are strengthened [6].

### 1.2. Emergency Preparedness in the Hospital Setting

Disasters caused by weather events commonly trigger the responses seen with other disasters such as violence and infectious disease outbreaks. Surge, evacuation, and shelter in place are the cornerstone of an all-hazards approach to disaster response. A hazard vulnerability assessment guides mitigation efforts to strengthen the physical infrastructure of a hospital to withstand expected threats. 

The US Department of Homeland Security [7] defines mitigation as actions that “reduce the loss of life and property by lessening the impact of future disasters” [7] (p. 5). Disaster preparation for hospital-based engineers and emergency preparedness planners focuses on mitigation, the completion of a hazards vulnerability assessment, and actions to strengthen and test the physical infrastructure of the hospital.

### 1.3. Disaster Training for Nurses

Disaster-related education for the pre-licensure nurse trainee is required by the American Association of Colleges of Nursing for doctoral, master’s, and baccalaureate nursing education [8,9]. Education for the hospital-based post licensure nurse is provided by their employer. Regardless of hospital type (e.g., public, private, non-profit, etc.), any hospital receiving federal reimbursement is held to the same emergency preparedness standards and enforced by accreditation bodies such as The Joint Commission (TJC) [10,11]. Emergency preparedness standards require hospitals to conduct two testing exercises annually based on their vulnerability assessment and establish a process for all staff to participate in the testing exercises over a period of time. Disaster competencies included amongst general nursing competencies are validated on hire and annually with large numbers of nurses moving through a range of skills stations [12]. Levine and Johnson describe learning modules as lacking interaction and a higher level of decision making that is often needed in emergency situations. 

Nonetheless, little is known about how hospital-based nurses in the United States perceive the disaster training they receive or feel about their readiness for disaster [13]. Instead, authors of several extant surveys describe small sample sizes, low population representation, and narrow geographic representativeness (e.g., rural), limiting the generalizability of their findings [14,15,16,17]. Surveyed nurses reported the lack of real-life disaster experience as the reason for lacking confidence in their abilities to respond [13,15,17,18]. 

### 1.4. Novel Approaches to Enhance Traditional Disaster Preparedness and Response

The complexity of recent disasters may suggest novel approaches to enhance traditional disaster preparedness activities. Despite participating in emergency planning activities to prepare for a surge response, hospitals described being unprepared as they faced significant challenges with staffing, supply chains, clinical space, and communication during the COVID-19 pandemic [19]. Hospitals resorted to a range of creative strategies to expand their capacity to manage patients and staff during the pandemic [20]. TJC considered questions and issues happening in real time from their accredited and critical access hospitals regarding crisis standards of care [21]. They observed that few health care systems had planned for the process of implementing crisis care strategies but instead focused on protocols for triaging specific interventions (e.g., ventilators). The experiences from the field drove new requirements, the elimination of redundancy, and the development of a concise and meaningful framework to support vital processes. Based on lessons learned from Hurricanes Harvey, Irma, and Maria, the National Academies of Sciences, Engineering, and Medicine [22] suggest augmenting the traditional focus of providing relief during a disaster with preemptive tactics to understand causes of unmet demand (e.g., identifying bottlenecks, gaps, and broken links in local supply chains) so stakeholders know how a hurricane can disrupt their supply chain and what they need to do to restore normal operations as soon as possible.

While mitigation is most often used as a term to strengthen the physical infrastructure of a hospital, frontline nurses could be considered as human infrastructure and the actions required of nurses during a disaster response are often the same activities performed every day, but with increased speed, intensity, and volume. Inefficiencies in patient flow are associated with poor patient care outcomes [23,24] and, left unaddressed, chronic inefficiencies (e.g., delays in care, supply chain issues, inadequate or lack of resources, etc.) are likely to worsen under the pressure of a disaster [25].

Lesson learned from all disasters, regardless of cause, serve to inform the response to disasters caused by the climate. Novel approaches to enhance traditional disaster preparedness and responses can only strengthen nurses, the human infrastructure within the hospital setting. The aim of this paper is to describe the subjective experience of nurse executives (NE) who have supervised nurses’ responses to major disasters. This paper will focus on strategies to support the nursing response to disaster, specifically to strengthen resiliency and the ability to maintain function despite the shock of a disaster, including those caused by climate change. 

This work represents efforts by the Veterans Emergency Management Evaluation Center (VEMEC) to support a 2014 Disaster Nursing Call to Action [26,27,28,29]. The Department of Veterans Affairs is the largest employer of nursing personnel in the United States (US) and has a vested interest in identifying organizational strategies to support nurses’ response to disaster events. 

## 2. Methods

The research team consists of five researchers, all with disaster experience. The study was reviewed and approved by the VA GLA IRB (IRB) using an expedited IRB review. All IRB approval criteria continue to be met in 38 CFR §16.111 and 21 CFR §56.111 for FDA-regulated research. The risk classification of the study continues to be minimal risk, and the documentation of informed consent was waived. The participants received a research information sheet covering all points of consent prior to the interview. 

All research team members were aware of potential bias due to their extensive experience with disasters and/or research with disaster survivors. None of the research team members knew the nurse executive respondents personally and they had no prior professional relationship with them. 

### Study Design

A phenomenological approach was used to describe the lived experience of NEs who supervised nurses during a major disaster in the hospital setting. A phenomenological research design provides a framework leading to the understanding of the individual experiences and how a phenomenon affects a person during an event [30]. The purpose was to examine actions by the nursing leader during an emergency event to extrapolate the best practices along with lessons learned. The phenomenon of nurse actions taken in disaster preparedness, response, and recovery was of key interest to the researchers. Scientific phenomenology aims at describing a general or typical essential structure based on descriptions of experiences from others [31,32]. Descriptive phenomenology, developed by Husserl, is described as people experiencing life to understand the essence of the experience. Descriptive phenomenological studies involve the steps of bracketing, intuiting, analyzing, and describing [33].

Each member of the research team has either experienced disasters or has worked with those who have survived disaster events. The researchers examined information and data through their own lenses of experience and assumptions, aware of potential bias, but were open to hearing what the transcribed words of each nurse meant [bracketing] [34]. Essentially, the researchers attempted to gain a better understanding of the lived experiences of these nurse participants by being immersed in their worlds during data collection and analysis and open to the meanings attributed to the phenomenon by those who shared their experiences [intuiting] [33]. The analysis phase was completed as the researchers extracted significant statements and categorized them. As a final, descriptive phase, the researchers defined the phenomenon by directly relating the competencies or actions taken versus those actions not taken, simply observed, or recommended for action at a future event [33].

All research team members were aware of potential bias due to their extensive experience with disasters and/or research with disaster survivors. The research team members did not have personal or professional relationships with the nurse executive respondents.

Major disasters caused by a range of disaster types and geographical locations were considered based on news reports, public after-action reports, and key informants. Contact information was obtained through the hospital website or telephone directory. Thirty NEs who had experienced disasters caused by fire (3), infectious disease (13), violence (7), and weather (7) received an email invitation, an IRB approved research information sheet, and a follow-up telephone call. Of those invited, 17 did not respond, 2 refused without a stated reason and 2 provided 2 referrals. The 11 who agreed to participate had experience supervising events caused by infectious disease (4), violence (5), and weather (4). The events occurred between 2014 and 2020 and the length of time between the disaster event and the interview date ranged from 6 months to 3.5 years. 

The interviews were conducted by telephone after verbal agreement from private and secure offices and lasted approximately 30–60 min. Interviews were audio recorded and professionally transcribed. All transcripts had a quality assurance check against the voice recording. 

The interview guide included four questions: (1) Describe your observations of the range of roles and responsibilities assumed by the nurses you supervised; (2) In recalling the knowledge, skills, and abilities required of nurses during the disaster you supervised, can you describe what you think best prepares nurses? (3) Based on your experience and observations, can you suggest organizational strategies for supporting nursing responses during a disaster in the hospital setting? 

All research team members participated in coding. Each member was assigned as the lead coder for a shared number of interviews. Using Excel, the lead tagged quotation chunks with emerging codes and grouped them under main theme headings as they emerged. The lead presented a summary in memo format to the team for group discussion and reconciliation of any disagreements. As interviews were reviewed, two designated team members managed a master Excel file and methodology tracking document. Codes and theme categories were eventually collapsed during group discussions. If needed, a short follow-up audio-recorded telephone call with the participant took place to verify that the data codes and theme categories were accurate (member checking). 

## 3. Results

Researchers reviewed all raw interview data codes and identified a subset representing a theme of applied mitigation to establish readiness. Table 1 illustrates the 11 NEs who participated in the study, their disaster experience, and presence of a theme subheading across all interviews.

### 3.1. Mitigation Steps to Establish Readiness

Assessment of human infrastructure: daily skills needed during disaster response;Identification and study of failure points and metrics;Strengthening human infrastructure: rectifying deficiencies;Monitoring metrics and making corrections during conventional periods.

Three of the NE experiences demonstrated theme saturation in these categories, and accordingly, this study focuses on those experiences to describe these applied mitigation steps that respondents identified as necessary to establish readiness. The experience of NE3 serves as a clear illustration of Mitigation Steps to Establish Readiness. NE3 looked back on a quality assurance project to address the failure point of holding patients in the emergency department and retrospectively realized the project served as a source of mitigation. The following are excerpts from the experience and demonstrate the concept of how everyday hospital strategies can serve as mitigation. The scale of incoming surge from a mass casualty incident (MCI) was compared to a plane crashing in the middle of a field. Victims arrived suddenly and in a disorganized fashion; few carried identification. 


*“They arrived by POV [privately owned vehicle], people carrying them on their shoulder, people carrying them in their arms, police cars, Uber drivers... they were literally loading people into the back of pickup trucks, as many as they could carry that were wounded, and they were bringing them into our organization.”*


Onsite doctors and nurses from every department worked in teams, as if they were in a “war” or “war zone”. They used the lawn of the hospital as an entry point to triage and transport patients to various parts of the hospital. Some patients went directly to the operating room (OR). Depending on the type of injury, unstable patients were taken to various intensive care units (ICUs) where they were intubated and stabilized until OR rooms became available (e.g., head trauma to neuro ICU, chest trauma to surgical ICU, etc.). Stable patients were taken to the post anesthesia care unit (PACU) to be medicated for pain while they waited for their turn to be taken into surgery. To make beds available, nurses in med/surg grouped several patients into one room, discharged those who were eligible, and transferred approximately 50 patients to “sister” hospitals. 


*“Within the course of about six hours, we had 212 patients present for treatment. Within about three hours, we had 80 or so of them for surgery.”*


### 3.2. Assess Human Infrastructure: Daily Skills Needed during Disaster Response

A year and a half before the MCI, the entire hospital examined a daily habit of holding admitted patients in the emergency department (ED) for extended periods while they waited for beds on the med/surge nursing units. This NE described a throughput exercise to decrease patients being held in the Emergency Department which ultimately prepared the nursing staff to respond to a mass casualty event. 


*“We weren’t preparing for a mass casualty incident. We were trying to become efficient taking care of 176,000 ER visits and deliver safe effective care. And we had relentlessly focused on that for a good year and a half.”*



*“...when the patients were coming to the floor [during the MCI], they were just doing it at a quicker clip than what we did on a day-to-day basis.”*


### 3.3. Identify and Study Failure Points and Metrics

This nurse executive set an expectation that every manager and their units agree to identify and study all delays between hospital admission and discharge to vehicle. 


*“We simply defined…our failure points and said holding 80 patients is a failure in our system and then began to follow the string back to where the processes were breaking down in the system. I mean we measured every data point you could measure and just worked on making them better with the individual stakeholders.”*


### 3.4. Strengthen Human Infrastructure: Rectify Deficiencies

Each delay was replaced with a more efficient behavior and practiced until holding patients in their ED was no longer a habit. Every nurse understood they were personally accountable and equally culpable for efficiently managing patient volume from the order to admit, to assisting patients into their transport vehicle at discharge, regardless of department. 


*“And we had relentlessly focused on that for a good year and a half, and so when the patients started coming in at that pace…imagine holding 80 to 110 patients admitted and still seeing 176,000 patients a year. They were doing that every day, day in and day out. And all we did to facilitate was throw the back door of the ED open and teach the inpatient units how to handle the flow.”*


### 3.5. Monitor Metrics and Make Corrections during Conventional Periods

The culture changed and the new behaviors became the norm. Nurses throughout the hospital were surprised to realize the impact their work habits had on the management of patient flow within the ED. 


*“And so of course I had a surge policy. So how I comprehended it was a surge policy—what I would do if we get to this number of patients, we get to this number of patients, we get to that number of patients. But what they’re asking—and honestly, I didn’t know what that meant until [the MCI event]. Now I understand what that meant. I feel like I’m a good CNO, above average in some ways, and I didn’t understand it. I would say that most CNOs don’t understand what they’re being asked. You’re only thinking about—we’re thinking about the more tangible things, like, where we put people, you know, where I’m going to overflow to, when it goes layers and layers and layers deeper than that.”*



*“I took a very broken hospital and took a few good leaders, and we changed the culture. And because we did that, a lot of people lived.”*


## 4. Discussion

The subjective experiences of nurse executives illustrate the magnitude of disruption when hospitals are impacted by disaster and the depth of strength required to respond. Assessing daily activities, studying failure points, strengthening, and rectifying weaknesses and monitoring during conventional times serves as a parallel to how emergency managers approach mitigation efforts to strengthen the physical structure of a hospital. Applying a mitigation mindset to nurses as the human infrastructure of the hospital may enhance traditional disaster content and drills and ultimately help to prepare for disasters as the impact of climate change continues. 

A mitigation mindset for nurses fits within existing regulatory requirements. Effective as of July 2022, TJC updated their emergency management standards to include ensuring “that critical components of the program are addressed in the mitigation, preparedness, response, and recovery phases and integrated throughout the organization and within the larger community response network” [35] (p. 2). Relatedly, the requirements promulgated by Centers for Medicare and Medicaid Services [10] require that “The comprehensive risk assessment should include all risks that could disrupt the facility’s operations and necessitate emergency response planning to address the risk mitigation”. 

Using a mitigation mindset to strengthen the human infrastructure within a hospital supports resiliency and the ability to maintain functions during a disaster. Assessing daily activities is consistent with familiar continuous quality improvement methodologies such as Lean Six Sigma [36], Plan-Do-Study-Act cycles [37], and the Baldrige criteria [38] but with a specific operational outcome: disaster preparedness. Identifying and studying failure points can happen during conventional times. Some activities might include measuring the speed and accuracy of blood or medication administration or the speed of completing diagnostic tests. The availability of life-saving equipment (e.g., personal protective equipment, narcotics, pharmaceutical drugs for intubation, chest tubes, etc.) could also be monitored and tested daily. Clinical practice guidelines [39,40] could provide guidance to rectify deficiencies. The ongoing monitoring of metrics during and adjacent to periods such as weekends, influenza season, staffing shortages, or security breach events could provide insight into failure points.

Furthermore, creating a culture of preparedness to withstand the worst disaster, as a part of the proposed mitigation framework, is consistent with the recommendations made by a National Academies of Sciences, Engineering, and Medicine [41] committee to “Establish and sustain a system-wide culture that promotes help-seeking behaviors and supports psychological safety” (p. 2) and supports the concept of creating a culture of preparedness to mitigate the damaging impacts of occupational and disaster exposure. 

The narrative example provided by NE3 is a valuable illustration of NAM’s [22] suggestion to understand the causes of disruption to know how to repair and restore normal operations. Staff identified bottlenecks, gaps, and broken links in their discharge process which gave them the knowledge and skills to manage the flow of surge after a mass casualty event and quickly restore normal operations. 

A culture of preparedness or a culture of safety depends on organizational services (e.g., wellness services provide by human resources, employee assistance programs, etc.) which are needed every day and during disaster responses. Assessing, strengthening, and testing those services during drills and conventional times can create or enhance a safety net capable of responding during and after a disaster. Workplace violence preventions standards [42] and training resources such as de-escalation training [43] serve as examples of guidance to rectify deficiencies [44]. 

## 5. Limitations

As a small qualitative study, the generalizability of our findings is limited. However, there were several themes related to preparedness/mitigation/education, response/education, and recovery that were common across the participants’ responses. The nurse executives emphasized the need to actively participate in mitigating potential damage during non-disaster periods to enhance resilience and to maintain essential functions during disasters. 

The concept and significance of recalling surprising, emotional, and consequential events has been named as a flashbulb memory [45,46]. Studies focused on the successful recall of flashbulb memories related to disaster events have largely focused on general populations. For example, one study found the retention of memories of the attacks on 11 September 2001, diminished within the first year and then stabilized for as long as ten years [47]. Another study found the consistency of memories about 11 September 2001, to be associated with the degree of experienced anxiety and covert rehearsal about the event [48]. Studies of memory recall amongst nurses after disaster events are not found. However, the emotional impact of disasters on nurses is sufficiently documented [49]. For this study, the time from the disaster event to the interview varied.

## 6. Conclusions

Nurses are predictably on the frontline of disaster response, especially in the hospital setting. Severe weather events brought about by climate change cause a threat to human health. This paper adds to the body of evidence supporting the augmentation of traditional training to include practical mitigation efforts. While mitigation is a familiar term used to describe strategies to harden the physical infrastructure of a hospital, nurse executives who have supervised nurses’ responses to major disasters in the hospital setting described strategies which strengthened and protected the human infrastructure within a hospital—the frontline nurses. Future research should examine any differences between NEs’ recall of events within three months of a disaster verses six months or longer post event. Mitigation steps may improve outcomes in hospital function during conventional times; therefore, these steps may improve resiliency and the ability to maintain functions during major disasters, including climate change. 

## Figures and Tables

**Table 1 ijerph-21-00221-t001:** Nurse executive disaster experience and theme saturation.

			Theme Saturation			
Nurse Executive	Disaster	Impact	1	2	3	4
NE1	Weather	Tornado destroyed hospital, required emergent evacuation	*			
NE2	Weather	Hurricane caused flooding and generator failure, required emergent evacuation				
NE3	Violence	Mass casualty incident, 212 critical surge victims, 75% dead on arrival	*	*	*	*
NE4	Violence	Mass casualty incident, 50 emergent surgical cases	*			
NE5	Violence	Active shooter on multiple nursing units, 2 fatalities, multiple injured		*	*	
NE6	Violence	Onsite active shooter in perioperative department, 2 fatalities, multiple injured	*	*	*	*
NE7	Weather	Hurricane caused flooding and destruction of hospital infrastructure, extended shelter in place followed by evacuation				
NE8	Infectious Disease	Influenza season, surge required alternate care sites				
NE9	Weather, Violence, Infectious Disease	Hurricane, shelter in placeBombing, critical care surge victimsMass casualty incident, critical care surge victims Ebola, presented to emergency department	*	*	*	*
NE10	Infectious Disease	COVID-19, surge	*			
NE11	Infectious Disease	COVID-19, alternate care site	*			

Mitigation Step to Establish Readiness: 1: Assess human infrastructure: daily skills needed during disaster response; 2: Identify and study failure points and metrics; 3: Strengthen human infrastructure: rectify deficiencies; 4: Monitor metrics and make corrections during conventional periods. * Indicates presence of a Mitigation Step to Establish Readiness.

## Data Availability

The data that support the findings of this study are not publicly available because the participants of this study did not provide written consent for their data to be shared publicly and they contain information that could compromise the privacy of research participants.

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
