# Peer review of "Culture of Safety and Preparedness: Benefits of Applying a Daily Mitigation Mindset in the Hospital Setting"

_ijerph, 2024, doi:10.3390/ijerph21020221_

Round 1

Reviewer 1 Report

Comments and Suggestions for Authors

This study described the subjective experience of nurse executives (NE) who have supervised nurses’ response to major disaster, focusing on strategies to support nursing response to disaster. In general, the study is well designed with logical writing and the authors described the story with empirical material, especially the in-depth interview. The comments are as follows:

1. In the introduction, the authors mentioned the requirements of PRIVATE SECTOR health leaders and emergency management partners in emergency preparedness and response initiatives to climate change. What about the requirements of PUBLIC SECTOR health leaders and emergency management partners? If the study has covered the health leaders and emergency management partners from public sectors, the introduction should also review the literature of these staff.

2. Based on above comments. Is it possible to provide more detailed information of the NEs, e.g., whether they are from private or private sectors, from community healthcare centers or major hospitals (levels of healthcare institutes), the names of states they are from? Thus, the representativeness of the NEs in the study can be better described and will also help authors in explaining limitations of the study.  

3. Tables 1 and 2 have overlapped contents, e.g., the disaster types. Is it possible to combine Table 1 with Table 2 and put in the results part? Thus, the authors may use a wide table covering more contents, plus the information suggested to add in above comment.

Comments on the Quality of English Language

The manuscript requires proofreading before acceptance for publication.

Reviewer 2 Report

Comments and Suggestions for Authors

Very interesting research, especially the rarely used method of phenomenological analysis (most often psychology and pedagogy). Therefore, I propose to write a few more sentences on this subject in the description of the research method. I also wonder about the time lag between the disaster experience and the interview: “The events occurred between 2014 and 2020, and the time between the disaster event and the interview date ranged from 6 months to 3.5 months.” Has there been a comparison of the results of interviews with people who had the longest and shortest periods of time since the disaster? I think that even if not, it is worth mentioning it and proposing further research steps in the conclusions, also in this aspect.
